# Zero-Dose Vaccination of Self-Paid Vaccines Among Migrant and Left-Behind Children in China: Evidence from Zhejiang and Henan Provinces

**DOI:** 10.3390/vaccines13020118

**Published:** 2025-01-24

**Authors:** Yaguan Zhou, Heng (Anna) Du, Shu Chen, Shenglan Tang, Xiaolin Xu

**Affiliations:** 1School of Public Health, The Second Affiliated Hospital, Zhejiang University School of Medicine, Hangzhou 310058, China; yaguan.zhou@zju.edu.cn; 2The Key Laboratory of Intelligent Preventive Medicine of Zhejiang Province, Hangzhou 310058, China; 3China Country Office, Bill & Melinda Gates Foundation, Beijing 100027, China; 4Duke Global Health Institute, Duke University, Durham, NC 27708, USA; 5Global Health Research Center, Duke Kunshan University, No. 8 Duke Avenue, Kunshan 215316, China; 6SingHealth Duke-NUS Global Health Institute, Duke-NUS (National University of Singapore), Singapore 119077, Singapore; 7School of Public Health, Faculty of Medicine, The University of Queensland, Brisbane 4153, Australia

**Keywords:** zero-dose vaccination, self-paid vaccines, vaccine equity, migrant children, left-behind children

## Abstract

Background/Objectives: As zero-dose vaccination has become a global health concern, understanding the practice of self-paid immunizations in migrant and left-behind children in China is crucial to the prevention and control of infectious diseases. Methods: A cross-sectional study was conducted in 1648 children and their caregivers in urban areas in Zhejiang Province and rural areas in Henan Province. The participants were then classified into four groups: urban local, migrant, non-left-behind, and left-behind. Results: Compared to urban local children, migrant (prevalence ratios: 1.29, 95% confidence intervals: 0.69–2.41), non-left-behind (4.72, 3.02–7.37), and left-behind (4.79, 3.03–7.56) children were more likely to be zero-dose vaccinated. Children aged 1–2 years (odds ratio: 1.60, 95% confidence intervals: 1.14–2.23) and born later (1.55, 1.12–2.14), with caregivers aged >35 years (1.49, 1.03–2.15) and less educated (elementary school or lower: 4.22, 2.39–7.45) were less likely to receive self-paid vaccinations, while caregivers other than parents (0.62, 0.41–0.94) and lower household income (0.67, 0.49–0.90) lowered the likelihood of zero-dose vaccination of self-paid vaccines. For migrant and rural zero-dose children, the majority of caregivers reported they “didn’t know where to get a vaccination”, with responses ranging from 82.3% to 93.8%. Conclusions: Migrant and rural children should be prioritized in the promotion of self-paid immunization in order to accomplish the WHO Immunization Agenda 2030’s goal of “leaving no one behind”.

## 1. Introduction

Since the establishment of the National Immunization Program (NIP) in 1978, China has made enormous progress in the control of infectious diseases [1], with over 16 million infants vaccinated each year [2]. Despite the strong recommendation of the World Health Organization (WHO), there are still a few vaccines to prevent infectious diseases that have not been included in the NIP in China [3]. These vaccines are voluntary and self-paid, contributing to wide vaccine inequity, with a considerable proportion of zero-dose children (receiving no vaccines) [4]. Over the past few decades, the substantial increase in population migration has generated two socioeconomically disadvantaged groups: migrant children in urban areas and left-behind children in rural areas in China [5,6]. In previous research, low self-paid vaccination rates were observed in these two growing groups of children [7]. To deliver targeted interventions for promoting the uptake of self-paid vaccines, more attention should be paid to the zero-dose vaccination population in migrant and left-behind families [8].

Therefore, this study aimed to analyze the rate and associated factors of zero-dose vaccination of self-paid vaccines and its potential factors among migrant and left-behind children.

## 2. Materials and Methods

### 2.1. Participants and Procedures

A cross-sectional survey was conducted in urban areas in Zhejiang Province, with a large number of migrant families, and rural areas in Henan Province, with a large number of left-behind families. A total of 1648 children aged 1–6 years and their caregivers were divided into four groups based on their migration status: urban local, migrant, non-left-behind, and left-behind. Left-behind families refer to one or both parents migrating into cities for work, leaving their children in the rural communities with other caregivers (e.g., grandparents) for over six months [7], while non-left-behind families refer to rural local counterparts. Migrant families refer to parents migrating into cities to work together with their children for over six months [9]. Zhejiang University School of Public Health Medicine Ethics Committees approved this study protocol (ZGL202206-6, 1 July 2022).

### 2.2. Measures

Self-paid vaccines: hemophilus influenza b (Hib), varicella, rotavirus, enterovirus 71 (EV71), and 13-valent pneumonia (PCV 13) vaccine [10].

Knowledge of vaccination: seven items, including convenience, category, efficiency, continuity, time, schedule, and adverse events. This variable was then dichotomized into good (aware of all items) and poor (not aware of at least one item).

Experience of vaccination: nine items, including convenience, reminder, environment, consultation, skills, service quality, process, education, and time. This variable was then dichotomized into satisfied (satisfied with all items) and unsatisfied (not satisfied with at least one item).

Demographic characteristics: the socio-demographic characteristics of the child included age (1–2 years vs. >2 years), sex (boy vs. girl), and birth order (first-born vs. later-born). The socio-demographic characteristics of the caregiver included family role (parents vs. others), age (≤35 years vs. >35 years), sex (male vs. female), education level (elementary school or lower vs. middle school vs. junior college or higher), total household income (less than average vs. more than average), physical health (assessed by 12-Item Short Form Survey [SF-12]), and mental health (assessed by SF-12).

### 2.3. Data Analysis

The zero-dose prevalence of self-paid vaccines, knowledge and experience of vaccination, socio-demographic characteristics, and reasons for the zero-dose vaccination of self-paid vaccines were described as a number (percentage).

Log-binomial regression models were used to calculate prevalence ratios (PRs) and 95% confidence intervals (CIs) for zero-dose vaccination in different types of children, regarding local urban children as a reference. To explore the associated factors of zero-dose vaccination of self-paid vaccines, socio-demographic characteristics were entered in a multivariable logistic regression model, and odds ratios (ORs) and 95% CIs were calculated to compare the association between the socio-demographic factors and zero-dose vaccination in the total sample and four migration types of children.

Analyses were performed using SAS 9.4 and R 3.6.1.

## 3. Results

The overall prevalence of zero-dose vaccination was 12.5% (206/1648) in the included families. Specifically, the prevalence was 4.3% (22/517) among urban local children, 5.8% (16/276) among migrant children, 19.7% (96/488) among non-left-behind children, and 19.6% (72/367) among left-behind children. Setting urban local children as the reference, migrant [prevalence ratios (PRs): 1.29, 95% confidence intervals (CIs): 0.69–2.41], non-left-behind (4.72, 3.02–7.37), and left-behind (4.79, 3.03–7.56) children were more likely to be zero-dose vaccinated (Figure 1). The zero-dose prevalence and PRs for single self-paid vaccines are shown in Table 1.

As for the associated factors, children aged 1–2 years [odds ratio (OR): 1.60, 95% CI: 1.14–2.23] and born later (1.55, 1.12–2.14), and caregivers aged >35 years (1.49, 1.03–2.15) and less educated (elementary school or lower: 4.22, 2.39–7.45) were less likely to receive self-paid vaccination. Caregivers other than parents (0.62, 0.41–0.94) and with lower household income (0.67, 0.49–0.90) was more likely to receive self-paid vaccination. However, the above-mentioned associations attenuated to non-significance in specific family types. For example, the higher likelihood of zero-dose vaccination was only observed among caregivers of urban local children with lower educational levels (middle school, 2.88, 1.10–7.54), and no characteristics significantly influenced the zero-dose vaccination in migrant children. For non-left-behind children, caregivers with poor knowledge of immunization (0.57, 0.32–0.99) were more likely to get their children vaccinated with ≥1 self-paid vaccine. For left-behind families, the later-born children experienced higher likelihood of zero-dose immunization (1.91, 1.09–3.36) (Table 2).

Table 3 shows the reasons for zero-dose vaccination of self-paid vaccines. For urban local zero-dose families, the main reasons for zero-dose vaccination were “never heard this vaccine” (40.9%) and “heard of negative information of vaccination” (31.08%). In the other three families, a large proportion of caregivers reported that they “didn’t know where to get a vaccination”, with responses ranging from 82.3% to 93.8%. In addition, nearly half of the caregivers of rural children reported the reasons “the efficacy of self-paid vaccines is concerning”, “having no time to get the children vaccinated”, and “there is no risk of contracting the disease”.

## 4. Discussion

The growing number of zero-dose children has been well recognized by global health strategies. For example, the WHO Immunization Agenda 2030 [11] and the Gavi Strategy 5.0 [12] set out the goal of “leaving no one behind” [9] and put core focus on reaching zero-dose children and increasing equitable use of vaccines. Although China has made many efforts to promote nationwide vaccination over decades, many children still do not receive any self-paid vaccines. This study not only observed a difference in zero-dose vaccination coverage between urban and rural areas but also found that the rates of zero-dose vaccination of urban migrant children, rural non-left-behind, and left-behind children were higher than those of urban local children. Compared to urban local children, the other three types of children are at a socioeconomic disadvantage, with poorer sanitation and lower utilization of health services [13,14]. Therefore, reducing or eliminating the equity on self-paid vaccination should be focused on migrant children and rural children.

For non-left-behind families, the probability of zero-dose vaccination was higher for families of children aged 1–2 years and caregivers aged >35 years. These findings suggested that self-paid vaccination promotion should be delivered as soon as possible for families with age-eligible children, and younger family members should be encouraged to participate in the decision-making process of self-paid vaccination. However, the knowledge level of immunization was reversely associated with zero-dose vaccination of self-paid vaccines, which may be due to the bias of information sources in rural areas [15]. For left-behind children, those born later were more likely to be unvaccinated than those born first. One possible explanation was that the first-born children gained more attention and support from family members, while those born later had limited access to parental time and supervision [16,17]. Caregivers who were not parents (mainly grandparents) and with lower household income were more likely to vaccinate their children, perhaps suggesting grandparent–grandchild cohesion theory, which refers to the intimate emotional bond between children and their grandparents [18]. It also highlights the need to consider the structure and socioeconomic status of left-behind families when policymakers and vaccinators implement non-NIP vaccination promotion programs. Flexible vaccination promotion programs, including mobile vaccination units or school-based vaccination programs, also have high practical value. Future studies are also needed to explore whether logistic factors, such as distance to healthcare facilities, availability of health professionals, or vaccination costs, impede self-paid vaccine uptake.

The reasons for zero-dose self-paid vaccination among the four children types were also analyzed. The main reasons included “don’t know where to get vaccinated”, “have no time”, “think no risk of contracting the disease”, and “concern about the efficacy”. These findings provided primary workers imperatives for standardizing the vaccination process and alternating vaccination service hours (e.g., off-peak services). Health education, including knowledge of target diseases, the safety and efficacy of each self-paid vaccine, and the local vaccination process, can help promote self-paid vaccination.

This study had some limitations. First, the cross-sectional design precluded us from exploring the longitudinal effects of the associated factors. Future longitudinal studies are warranted to offer insights into how associated factors influence zero-dose vaccination over time. Second, this study only focused on self-paid vaccines. Including data on government-provided vaccines would present a more comprehensive picture of vaccine equity, particularly in vulnerable populations. Third, our findings should be generalized with caution because the study participants were from two provinces in China. However, these two provinces are the major labor-importing or labor-exporting provinces in China, and the simple random sampling method was used to select participants. Therefore, serious bias is impossible. Fourth, due to the small sample size, the sample of some categories of variables were too small (e.g., age and birth order of children and caregivers) to further conduct subgroup analyses. Future large studies focusing on those aged 1–2 years and born later are warranted to yield targeted intervention strategies.

## 5. Conclusions

The rate of zero-dose self-paid vaccination was higher among urban migrant, non-left-behind, and left-behind children in rural areas than among urban local children. The socio-demographic characteristics of children and caregivers (e.g., educational level) were associated with zero-dose self-paid vaccination, and caregivers reported their lack of time, information, and knowledge on self-paid vaccines. Migrant children and rural children should be optimized in the promotion of self-paid vaccination (e.g., targeted health education) to achieve the goal of “leaving no one behind”.

## Figures and Tables

**Figure 1 vaccines-13-00118-f001:**
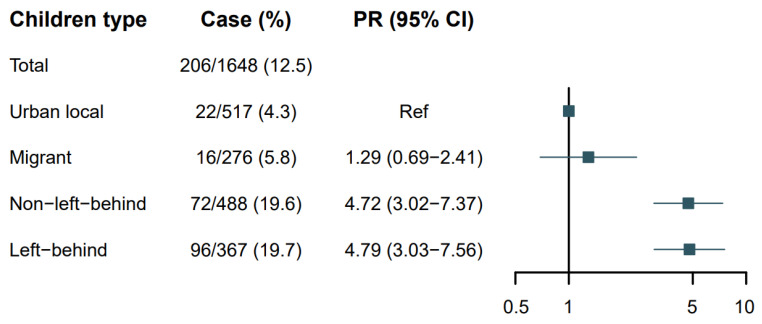
The zero-dose prevalence ratios (PRs) and 95% confidence intervals (CIs) of self-paid vaccines according to children type (n = 1648).

**Table 1 vaccines-13-00118-t001:** The zero-dose prevalence ratios (PRs) and 95% confidence intervals (CIs) of five self-paid vaccines according to children type (n = 1648).

Vaccine Type	Total (n = 1648)	Urban Local (n = 517)		Migrant (n = 276)		Non-Left-Behind (n = 488)		Left-Behind (n = 367)	
	Case (%)	Case (%)	PR (95% CI) ^1^	Case (%)	PR (95% CI)	Case (%)	PR (95% CI)	Case (%)	PR (95% CI)
Hib vaccine	566 (34.3)	125 (24.2)	Ref	149 (54.0)	2.13 (1.77–2.58)	156 (32.0)	1.34 (1.09–1.63)	136 (37.1)	1.54 (1.26–1.89)
Varicella vaccine	489 (29.7)	74 (14.3)	Ref	46 (16.7)	1.08 (0.77–1.52)	207 (42.4)	2.95 (2.34–3.72)	162 (44.1)	3.02 (2.38–3.82)
Rotavirus vaccine	1005 (61.0)	190 (36.8)	Ref	156 (56.5)	1.53 (1.31–1.79)	354 (72.5)	1.99 (1.75–2.25)	305 (83.1)	2.25 (1.99–2.54)
EV71	827 (50.2)	201 (38.9)	Ref	127 (46.0)	1.16 (0.98–1.37)	285 (58.4)	1.52 (1.33–1.73)	214 (58.3)	1.50 (1.31–1.72)
PCV13	1078 (65.4)	187 (36.2)	Ref	142 (51.5)	1.42 (1.21–1.67)	412 (84.4)	2.34 (2.07–2.64)	337 (91.8)	2.54 (2.26–2.86)

^1^ The models were adjusted for the age and sex of children. PR, prevalence ratio; CI, confidence interval; Hib, Hemophilus influenza b; EV71, enterovirus 71 vaccine; PCV13, 13-valent pneumonia vaccine.

**Table 2 vaccines-13-00118-t002:** The associations between socio-demographic characteristics of children and caregivers, and zero-dose vaccination of self-paid vaccines among the four children types.

	n (%)	OR (95% CI)				
		Total (n = 1648)	Urban Local (n = 517)	Migrant (n = 276)	Non-Left-Behind (n = 488)	Left-Behind (n = 367)
Children						
Age (Years)						
1–2	406 (24.6)	1.60 (1.14–2.23)	1.23 (0.47–3.21)	1.19 (0.42–3.36)	2.03 (1.20–3.43)	1.80 (0.91–3.54)
>2	1242 (75.4)	Ref	Ref	Ref	Ref	Ref
Sex						
Boy	859 (52.1)	Ref	Ref	Ref	Ref	Ref
Girl	789 (47.9)	0.90 (0.67–1.22)	1.41 (0.59–3.38)	0.81 (0.28–2.34)	0.75 (0.46–1.20)	0.98 (0.57–1.71)
Birth order						
First-born	671 (40.8)	Ref	Ref	Ref	Ref	Ref
Later-born	974 (59.2)	1.55 (1.12–2.14)	1.39 (0.54–3.57)	0.73 (0.25–2.16)	1.52 (0.88–2.63)	1.91 (1.09–3.36)
Caregivers						
Family role						
Parents	1112 (67.5)	Ref	Ref	Ref	Ref	Ref
Others	536 (32.5)	0.62 (0.41–0.94)	1.84 (0.76–4.47)	0.33 (0.04–2.79)	0.68 (0.29–1.60)	0.36 (0.16–0.83)
Age (Years)						
<=35	1056 (64.1)	Ref	Ref	Ref	Ref	Ref
>35	592 (35.9)	1.49 (1.03–2.15)	0.92 (0.35–2.44)	1.23 (0.24–6.42)	2.36 (1.38–4.04)	1.44 (0.60–3.46)
Sex						
Male	287 (17.4)	1.11 (0.74–1.65)	1.00 (0.32–3.13)	0.70 (0.14–3.65)	0.90 (0.46–1.74)	1.26 (0.63–2.52)
Female	1361 (82.6)	Ref	Ref	Ref	Ref	Ref
Educational level						
Elementary school or lower	255 (15.5)	4.22 (2.39–7.45)	-	-	1.11 (0.36–3.41)	0.75 (0.11–5.16)
Middle school	863 (52.4)	3.03 (1.95–4.72)	2.88 (1.10–7.54)	0.53 (0.14–1.94)	0.80 (0.32–2.02)	0.75 (0.12–4.56)
Junior college or higher	530 (32.2)	Ref	Ref	Ref	Ref	Ref
Total household income ^1^						
Less than average	977 (59.3)	0.67 (0.49–0.90)	0.87 (0.34–2.19)	1.49 (0.39–5.75)	1.39 (0.85–2.28)	0.41 (0.23–0.75)
More than average	671 (40.7)	Ref	Ref	Ref	Ref	Ref
Physical health score ^2^						
Less than median	1080 (65.5)	0.90 (0.65–1.25)	1.66 (0.36–7.54)	0.59 (0.20–1.72)	1.32 (0.72–2.42)	0.83 (0.45–1.53)
Higher than median	568 (34.5)	Ref	Ref	Ref	Ref	Ref
Mental health score ^2^						
Less than median	1558 (94.5)	1.11 (0.51–2.38)	1.17 (0.15–9.03)	-	0.38 (0.10–1.48)	0.70 (0.13–3.74)
Higher than median	90 (5.5)	Ref	Ref	Ref	Ref	Ref
Knowledge of vaccination ^3^						
Poor	1234 (74.9)	0.97 (0.67–1.42)	1.30 (0.51–3.32)	1.39 (0.37–5.30)	0.57 (0.32–0.99)	1.46 (0.59–3.61)
Good	414 (25.1)	Ref	Ref	Ref	Ref	Ref
Experience of vaccination ^4^						
Unsatisfied	238 (14.4)	0.77 (0.48–1.22)	1.26 (0.44–3.61)	0.59 (0.12–2.84)	1.23 (0.64–2.36)	0.17 (0.02–1.35)
Satisfied	1410 (85.6)	Ref	Ref	Ref	Ref	Ref

OR, odds ratio; CI, confidence interval. Five multivariate logistic regression models were conducted for the whole sample and urban local, migrant, non-left-behind, and left-behind children. All the above characteristics of children and caregivers were entered into each model. ^1^ The total household income was measured by the sum of earning income, capital income, pension income, income from government transfers, other income, and total income from other household members during last year and was Chinese Yuan (CNY) 300,000, CNY 200,000, CNY 40,000, and CNY 40,000 for local urban, migrant, non-left-behind, and left-behind children, respectively. ^2^ The physical health and mental health of caregivers were assessed by the 12-Item Short Form Survey. ^3^ The knowledge of vaccination was measured using seven items: convenience, category, efficiency, continuity, time, schedule, and adverse events. This variable was then dichotomized into good (aware of all items) and poor (not aware of at least one item). ^4^ The satisfaction of vaccination was measured using nine items: convenience, reminder, environment, consultation, skills, service quality, process, education, and time. This variable was then dichotomized into satisfied (satisfied with all items) and unsatisfied (not satisfied with at least one item).

**Table 3 vaccines-13-00118-t003:** Reasons for the zero-dose vaccination of self-paid vaccines among the four children types.

	Total (n = 206)	Urban Local (n = 22)	Migrant (n = 16)	Non-Left-Behind (n = 96)	Left-Behind (n = 72)	*p*
Never heard	29 (14.1%)	9 (40.9%)	3 (18.8%)	12 (12.5%)	5 (6.9%)	<0.001
Illness of the child	39 (18.9%)	4 (18.2%)	5 (31.3%)	23 (24.0%)	7 (9.7%)	0.067
No risk of contracting the disease	87 (42.2%)	5 (22.7%)	2 (12.5%)	46 (47.9%)	34 (47.2%)	0.011
Bad experience of vaccination	6 (2.9%)	2 (9.1%)	1 (6.3%)	2 (2.1%)	1 (1.4%)	0.219
Received the same type of vaccines	39 (18.9%)	5 (22.7%)	3 (18.8%)	20 (20.8%)	11 (15.3%)	0.787
Concern about safety	34 (16.5%)	3 (13.6%)	2 (12.5%)	13 (13.5%)	16 (22.2%)	0.451
Concern about efficiency	99 (48.1%)	4 (18.2%)	4 (25.0%)	46 (47.9%)	45 (62.5%)	<0.001
Too many doses need to vaccinate	13 (6.3%)	1 (4.5%)	1 (6.3%)	5 (5.2%)	6 (8.3%)	0.847
Insufficient supply of vaccines	10 (4.9%)	2 (9.1%)	0 (0.0%)	4 (4.2%)	4 (5.6%)	0.605
Don’t know where to get vaccinated	161 (78.2%)	5 (22.7%)	15 (93.8%)	79 (82.3%)	62 (86.1%)	<0.001
Limited service time	49 (23.8%)	5 (22.7%)	2 (12.5%)	26 (27.1%)	16 (22.2%)	0.613
Heard of negative information about vaccination	47 (22.8%)	7 (31.8%)	3 (18.8%)	17 (17.7%)	20 (27.8%)	0.309
Others don’t suggest vaccination/to get a vaccination	39 (18.9%)	0 (0.0%)	6 (37.5%)	26 (27.1%)	7 (9.7%)	<0.001
Have no time to get a vaccination	97 (47.1%)	5 (22.7%)	5 (31.3%)	56 (58.3%)	31 (43.1%)	0.007

## Data Availability

The datasets used in this study are available from the corresponding author upon reasonable request.

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
