# Peer review of "Zero-Dose Vaccination of Self-Paid Vaccines Among Migrant and Left-Behind Children in China: Evidence from Zhejiang and Henan Provinces"

_vaccines, 2025, doi:10.3390/vaccines13020118_

Round 1

Reviewer 1 Report

Comments and Suggestions for Authors

In the manuscript by Zhou et al, the authors present data pertaining to non-mandatory vaccine uptake in children (from two different provinces of China).  The data collected included comparing children from migrant and local populations, children who were left-behind vs. those non-left behind, as well as other factors including the order of birth, and the age of the caregivers.

Overall, this is a poorly written paper, with the quality of English being so poor that it was difficult for the Reviewer to completely understand the manuscript.  Furthermore, the data was, for the most part, poorly presented.  Instead of simply giving of the raw data with their associated stats/confidence intervals, the authors should have presented all of their data in a manner similar to that presented in Figure 1.  The authors would also be well-advised to provide a clear definition for each group.

In it's current form, this manuscript is not fit for publication. 

Comments on the Quality of English Language

It was really bad.

Author Response

In the manuscript by Zhou et al, the authors present data pertaining to non-mandatory vaccine uptake in children (from two different provinces of China).  The data collected included comparing children from migrant and local populations, children who were left-behind vs. those non-left behind, as well as other factors including the order of birth, and the age of the caregivers.

Overall, this is a poorly written paper, with the quality of English being so poor that it was difficult for the Reviewer to completely understand the manuscript.  Furthermore, the data was, for the most part, poorly presented.  Instead of simply giving of the raw data with their associated stats/confidence intervals, the authors should have presented all of their data in a manner similar to that presented in Figure 1. The authors would also be well-advised to provide a clear definition for each group.

In it’s current form, this manuscript is not fit for publication.

Response: Thank you for your time and feedback. We have revised the full manuscript to improve clarity, particularly the language editing. First, we think that the formats of Table 1 and 2 are appropriate. As the results were reported according to four children types, there would be many columns if presented in a manner similar to Figure 1. Therefore we have kept the original tables in the current manuscript. Second, to make this study easier to understand, we added definitions of all four categories of families in 2.1. Participants and procedures section. In brief, left-behind families refer to one or both parents migrating into cities for work, leaving their children in the rural communities with other caregivers for over six months, while non-left-behind families refer to rural local counterparts. Migrant families refer to parents migrating into cities to work together with their children for over six months. We also provide references in the text (National Bureau of Statistics, 2021; Zhou, Y. et al, 2023).

In the 2.1. Participants and procedures section (Page 2, Lines 58-62): “Left-behind families refer to one or both parents migrating into cities for work, leaving their children in the rural communities with other caregivers (e.g., grandparents) for over six months, while non-left-behind families refer to rural local counterparts. Migrant families refer to parents migrating into cities to work together with their children for over six months.”

Reviewer 2 Report

Comments and Suggestions for Authors

The paper is excellently written and highlights the public health problem of inadequate self-paid vaccination in migrant children, left-behind children and non-left-behind children in China.

The researchers conducted an in-depth study on the association between vaccination with vaccines that are not part of the national vaccination program in China but are recommended and must be paid for in full. As a result, the expected differences in vaccination coverage occur depending on financial capabilities, parental education and the environment in which the child lives.

However, I suggest that for clarification, the authors add definitions of all four categories of children (urban local, migrant, non-left-behind and left-behind) in section 2.1. Participants and procedures in order to make the socio-demographic status of children and its consequences more understandable to readers who are not familiar with the challenges of vaccination in migrant and left-behind populations.

Author Response

The paper is excellently written and highlights the public health problem of inadequate self-paid vaccination in migrant children, left-behind children and non-left-behind children in China.

The researchers conducted an in-depth study on the association between vaccination with vaccines that are not part of the national vaccination program in China but are recommended and must be paid for in full. As a result, the expected differences in vaccination coverage occur depending on financial capabilities, parental education and the environment in which the child lives.

However, I suggest that for clarification, the authors add definitions of all four categories of children (urban local, migrant, non-left-behind and left-behind) in section 2.1. Participants and procedures in order to make the socio-demographic status of children and its consequences more understandable to readers who are not familiar with the challenges of vaccination in migrant and left-behind populations.

Response: Thank you for your time and useful comments. We have revised the paper to address your concerns and to include your suggestions.

To make this study easier to understand, we added definitions of all four categories of families in 2.1. Participants and procedures section. In brief, left-behind families refer to one or both parents migrating into cities for work, leaving their children in the rural communities with other caregivers for over six months, while non-left-behind families refer to rural local counterparts. Migrant families refer to parents migrating into cities to work together with their children for over six months. We also provide references in the text (National Bureau of Statistics, 2021; Zhou, Y. et al, 2023).

In the 2.1. Participants and procedures section (Page 2, Lines 58-62): “Left-behind families refer to one or both parents migrating into cities for work, leaving their children in the rural communities with other caregivers (e.g., grandparents) for over six months, while non-left-behind families refer to rural local counterparts. Migrant families refer to parents migrating into cities to work together with their children for over six months.”

Reviewer 3 Report

Comments and Suggestions for Authors

Introduction

The study addresses a pressing public health concern: zero-dose vaccination among migrant and left-behind children in China. By highlighting disparities in self-paid vaccine uptake, the authors provide critical insights into how social and economic factors influence vaccination rates. With global attention on achieving equitable vaccine access, the findings of this study align with the WHO Immunization Agenda 2030, emphasizing the need to "leave no one behind." This review critically evaluates the study's strengths and identifies opportunities for improvement.

Strengths

Relevance to Global Health Goals

The study is highly relevant to contemporary public health discussions. Zero-dose vaccination has become a global issue, and understanding its determinants in vulnerable populations such as migrant and left-behind children in China contributes to global efforts toward universal immunization.

Robust Study Design

The cross-sectional design, involving 1,648 children and their caregivers across diverse settings in Zhejiang and Henan provinces, ensures the study captures a wide spectrum of socio-demographic contexts. The classification of participants into four distinct groups—urban local, migrant, non-left-behind, and left-behind—enables nuanced analyses.

Clear Statistical Insights

The study employs sound statistical measures, such as prevalence ratios and odds ratios with confidence intervals, to provide evidence-based conclusions. The detailed breakdown of risk factors (e.g., caregiver education, child birth order) and protective factors (e.g., caregiver identity, household income) enhances the depth of the findings.

Identification of Key Barriers

A striking finding is the high proportion of caregivers who "didn’t know where to get vaccination," particularly in migrant and rural populations. This highlights an actionable area for public health interventions, emphasizing the need for awareness campaigns and improved healthcare accessibility.

Alignment with Policy Frameworks

The study's conclusion aligns with the WHO Immunization Agenda 2030, advocating for prioritization of migrant and rural children in vaccination promotion. This provides a policy-relevant takeaway that can guide future strategies.

Opportunities for Improvement

Addressing Causality

While the cross-sectional design provides valuable associations, it limits the ability to infer causality. Longitudinal studies could offer insights into how migration status, household income, and caregiver education dynamically influence vaccination uptake over time.

Expanding the Scope of Analysis

The study focuses predominantly on self-paid vaccines. Including data on government-provided vaccines would present a more comprehensive picture of zero-dose vaccination trends, particularly as these vaccines often target vulnerable populations.

Geographical and Cultural Representation

The study was conducted in two provinces, Zhejiang and Henan, which, while diverse, may not fully capture the broader cultural, economic, and healthcare disparities across China. Expanding the study to include more regions could improve generalizability.

Exploration of Healthcare Accessibility

Although the study identifies lack of knowledge about vaccination sites as a major barrier, it does not delve deeply into systemic healthcare access issues. Future studies could explore whether structural factors, such as distance to healthcare facilities, availability of health professionals, or vaccination costs, further impede vaccine uptake.

Qualitative Data Integration

While quantitative analyses are strong, the inclusion of qualitative data—such as interviews with caregivers—could provide richer contextual understanding of barriers to vaccination, particularly around cultural or social stigmas.

Policy Recommendations

The study concludes with a call for prioritizing migrant and rural children but stops short of providing specific actionable steps. More detailed policy recommendations, such as implementing mobile vaccination units or school-based vaccination programs, would strengthen its practical applicability.

Focus on Subpopulations

The findings indicate that younger children (aged 1–2 years) and later-born children are at higher risk of being zero-dose vaccinated. Further exploration of these subpopulations could yield targeted intervention strategies.

Conclusion

This study sheds light on an underexplored area of vaccine equity by focusing on self-paid vaccination uptake among migrant and left-behind children in China. Its findings underscore the urgent need for targeted interventions to improve vaccination rates in these vulnerable groups. However, addressing the limitations—such as the lack of causality, broader geographical representation, and deeper exploration of systemic barriers—could further enhance the study’s impact. Overall, this research provides a valuable foundation for informing policies aimed at achieving equitable immunization coverage and advancing global health goals.

Author Response

The study addresses a pressing public health concern: zero-dose vaccination among migrant and left-behind children in China. By highlighting disparities in self-paid vaccine uptake, the authors provide critical insights into how social and economic factors influence vaccination rates. With global attention on achieving equitable vaccine access, the findings of this study align with the WHO Immunization Agenda 2030, emphasizing the need to "leave no one behind." This review critically evaluates the study's strengths and identifies opportunities for improvement.

Response: Thank you for your time and positive feedback. We have revised the paper to include your comments and suggestions.

Strengths

Relevance to Global Health Goals: The study is highly relevant to contemporary public health discussions. Zero-dose vaccination has become a global issue, and understanding its determinants in vulnerable populations such as migrant and left-behind children in China contributes to global efforts toward universal immunization.

Robust Study Design: The cross-sectional design, involving 1,648 children and their caregivers across diverse settings in Zhejiang and Henan provinces, ensures the study captures a wide spectrum of socio-demographic contexts. The classification of participants into four distinct groupsurban local, migrant, non-left-behind, and left-behindenables nuanced analyses.

Clear Statistical Insights: The study employs sound statistical measures, such as prevalence ratios and odds ratios with confidence intervals, to provide evidence-based conclusions. The detailed breakdown of risk factors (e.g., caregiver education, child birth order) and protective factors (e.g., caregiver identity, household income) enhances the depth of the findings.

Identification of Key Barriers: A striking finding is the high proportion of caregivers who "didn’t know where to get vaccination," particularly in migrant and rural populations. This highlights an actionable area for public health interventions, emphasizing the need for awareness campaigns and improved healthcare accessibility.

Alignment with Policy Frameworks: The study's conclusion aligns with the WHO Immunization Agenda 2030, advocating for prioritization of migrant and rural children in vaccination promotion. This provides a policy-relevant takeaway that can guide future strategies.

Response: Thank you for your time and positive feedback.

Opportunities for Improvement

  1. Addressing Causality: While the cross-sectional design provides valuable associations, it limits the ability to infer causality. Longitudinal studies could offer insights into how migration status, household income, and caregiver education dynamically influence vaccination uptake over time.

Response: We agree with the reviewer that cross-sectional study design limits the ability to infer causality between associated factors and zero-dose self-paid vaccination. Future longitudinal studies are warranted to offer insights into how associated factors influence zero-dose vaccination over time. The limitation has been mentioned in the Discussion section.

In the 4. Discussion section (Page 7, Lines 192-195): “First, the cross-sectional design precluded us from exploring the longitudinal effects of the associated factors. Future longitudinal studies are warranted to offer insights into how associated factors influence zero-dose vaccination over time.”

  1. Expanding the Scope of Analysis: The study focuses predominantly on self-paid vaccines. Including data on government-provided vaccines would present a more comprehensive picture of zero-dose vaccination trends, particularly as these vaccines often target vulnerable populations.

Response: We agree with the reviewer that government-provided vaccines also need attention to promote the immunization equity in China. However, as the government-provided vaccines are free and mandatory, the phenomenon of zero-dose vaccination is sporadic. Therefore, we only focused on the zero-dose children of self-paid vaccines to address the wide vaccine inequity. We have deeply discussed this limitation in the Discussion section.

In the 4. Discussion section (Page 7, Lines 195-197): “Second, this study only focused on self-paid vaccines. Including data on government-provided vaccines would present a more comprehensive picture of vaccines equity, particularly in vulnerable populations.”

  1. Geographical and Cultural Representation: The study was conducted in two provinces, Zhejiang and Henan, which, while diverse, may not fully capture the broader cultural, economic, and healthcare disparities across China. Expanding the study to include more regions could improve generalizability.

Response: We agree with the reviewer that sampling in only two provinces would damage the data representativeness. However, Zhejiang Province is the major labor-importing province in China, and Henan Province is the major labor-exporting province in China. Therefore, there exists a large number of migrant families or left-behind families. By using the simple random sampling method, five towns/communities were selected in Zhejiang and Henan Provinces respectively. Therefore, serious selection bias is impossible.

In the 4. Discussion section (Page 7, Lines 197-201): “Third, our findings should be generalized with caution because study participants were from two provinces in China. However, these two provinces are the major la-bor-importing or labor-exporting province in China, and simple random sampling method was used to select participant. Therefore, serious bias is impossible.”

  1. Exploration of Healthcare Accessibility: Although the study identifies lack of knowledge about vaccination sites as a major barrier, it does not delve deeply into systemic healthcare access issues. Future studies could explore whether structural factors, such as distance to healthcare facilities, availability of health professionals, or vaccination costs, further impede vaccine uptake.

Response: Thanks for your suggestions. We have added future expectations in this field.

In the 4. Discussion section (Page 6, Lines 182-184): “Future studies are also needed to explore whether logistic factors, such as distance to healthcare facilities, availability of health professionals, or vaccination costs, impede self-paid vaccine uptake.”

  1. Qualitative Data Integration: While quantitative analyses are strong, the inclusion of qualitative datasuch as interviews with caregiverscould provide richer contextual understanding of barriers to vaccination, particularly around cultural or social stigmas.

Response: Thanks for your suggestions while it appears to be beyond the scope of our current research focus. Therefore, we are unable to make the requested modifications at this time.

  1. Policy Recommendations: The study concludes with a call for prioritizing migrant and rural children but stops short of providing specific actionable steps. More detailed policy recommendations, such as implementing mobile vaccination units or school-based vaccination programs, would strengthen its practical applicability.

Response: Thanks for your suggestions. We have added relevant statements in the revised manuscript.

In the 4. Discussion section (Page 6, Lines 180-182): “Flexible vaccination promotion programs, including mobile vaccination units or school-based vaccination programs, also have high practical value.”

  1. Focus on Subpopulations: The findings indicate that younger children (aged 12 years) and later-born children are at higher risk of being zero-dose vaccinated. Further exploration of these subpopulations could yield targeted intervention strategies.

Response: Thanks for your suggestions. However, we are unable to make the requested modifications at this time due to the small sample size. We have deeply discussed this limitation in the Discussion section.

In the 4. Discussion section (Page 7, Lines 201-204): “Fourth, due to the small sample size, the sample of some categories of variables were too small (e.g., age and birth order of children and caregivers) to further conduct subgroup analyses. Future large studies focusing on those aged 1-2 years and born later are warranted to yield targeted intervention strategies.”

Conclusion

This study sheds light on an underexplored area of vaccine equity by focusing on self-paid vaccination uptake among migrant and left-behind children in China. Its findings underscore the urgent need for targeted interventions to improve vaccination rates in these vulnerable groups. However, addressing the limitationssuch as the lack of causality, broader geographical representation, and deeper exploration of systemic barrierscould further enhance the study’s impact. Overall, this research provides a valuable foundation for informing policies aimed at achieving equitable immunization coverage and advancing global health goals.

Response: Thank you for your time and positive feedback. We have revised the paper to include your suggestions.

Round 2

Reviewer 3 Report

Comments and Suggestions for Authors

The authors have performed the changes as recommended. The manuscript can be accepted for publication.